# Mapping Evidence on the Epidemiology and Cost Associated with Maxillofacial Injury among Adults in Sub-Saharan Africa: A Scoping Review Protocol

**DOI:** 10.3390/ijerph20021531

**Published:** 2023-01-14

**Authors:** Adekunle I. Adeleke, Mbuzeleni Hlongwa, Sizwe Makhunga, Themba G. Ginindza

**Affiliations:** 1Discipline of Public Health Medicine, School of Nursing and Public Health, University of KwaZulu-Natal, Durban 4041, South Africa; 2Burden of Disease Research Unit, South African Medical Research Council, Cape Town 7925, South Africa; 3Cancer & Infectious Diseases Epidemiology Research Unit (CIDERU), College of Health Sciences, University of KwaZulu-Natal, Durban 4041, South Africa

**Keywords:** maxillofacial injury/facial trauma, epidemiology, risk factors, cost, sub-Saharan Africa

## Abstract

(1) Background: Maxillofacial injury (MI) occurs universally, for it disregards preference for age, gender, and geographical region. The global incidence and prevalence of facial fractures rose by 39.45% and 54.39%, respectively, between the years 1990 to 2017. Projections indicate that the burden of injuries will persist in sub-Saharan Africa (SSA) in the next twenty years. This scoping review aims to map the literature on MI epidemiology and the economic burden on society in SSA. (2) Methods: The methodology presented by Arksey and O’Malley and extended by Levac and colleagues will be employed in the scoping review. The researcher will report the proposed review through the Preferred Reporting Items for Systematic Review, and Meta-Analysis extension for scoping reviews (PRISMA-ScR). The review will include studies encompassing MI in sub-Saharan African adults 18 years and above. (3) Results: This will be presented as a thematic analysis of the data extracted from the included studies, and the Nvivo version 12 will be employed. (4) Discussion: We anticipate searching for related literature on the prevalence, incidence, risk factors, mortality, and cost associated with MI in the adult population of SSA. The conclusion from the review will assist in ascertaining research gaps, informing policy, planning, authorizing upcoming research, and prioritizing funding for injury prevention and management.

## 1. Introduction

Evidence has shown that injuries account for a significant health burden in many countries annually [1,2,3]. The World Health Organisation (WHO) report of 2019 revealed that injuries resulted in more than 4.3 million deaths annually, translating to 11,780 deaths per day [4]. Markedly, low- and middle-income countries (LMICs) accounted for 89% of the deaths, while 10% were attributed to sub-Saharan Africa (SSA) [5,6]. Maxillofacial injury (MI) is categorized as a nature of injury [6] and is characterized by the injury that may result in damage to soft and hard tissues of the facial region in the form of burns, lacerations, and fractures [7,8]. The fractures that entail the orbits, maxillary, nasal, mandible, and other bones in the facial region [9,10,11]. The incidence of fracture of the facial bones in 2019 was estimated at 7.5 million, with 1.8 million persons existing with the disability and this resulted in 117,402 years lived with a disability (YLDs) [6]. These parameters are significantly high in SSA [6].

Studies have identified the commonest mechanisms of injury in the maxillofacial region as; motor vehicle and motorcycle collisions [12,13,14], assault [8,15,16], and falls [17,18]. Additionally, MI has been identified as one of the commonest forms of injury in gender-based violence [19,20,21]. Maxillofacial injury occurs ubiquitously and has been associated with various risk factors [15,22] such as; demographic characteristics (age, gender) [8,23], substance abuse [24,25], social determinants of health (poverty, neighbourhood, and built environment) [26,27]; economic and gender inequalities [21,28,29]; frail criminal justice [30] and poor regulation of firearms [31]. The fracture in MI can exist as singular trauma to the orofacial region but often co-exists with other forms of trauma to the patient’s body and has been a significant public health problem globally [32,33]. Maxillofacial injury may seldom be life-threatening but may result in facial distortion and impaired occlusal support, which may require prosthetic rehabilitation [34] in restoring occlusal function, to avoid temporomandibular joint (TMJ) dysfunction [35,36]. However, MI may become critical when vital sensory organs, the upper airway, and the oro-digestive region are affected [37]. That might result in death or inconvertible sequelae such as intracranial injury and severe psychosocial disorders [38,39]. Maxillofacial injury with a traumatic head injury has been reported to be a global public health issue [33,40,41]. 

The higher ratio of injury YLDs in LMICs is yet to be translated to the investigations of the health burden associated with this nature of injury [22]. This disproportionality in association results from the lack of a functional injury surveillance system and the weakness in the medical certification of causes of death as per international classifications [42]. The economic burden emanates from the management and rehabilitation associated with restoring aesthetic, physical, and functional damage [12,33] on the individual and the healthcare system [16,33,43]. The average cost for managing combined maxillofacial traumas per patient ranges from $1716 to $24,004 in Turkey and Malaysia, respectively [33,44]. In Nigeria (Kano State), the average cost of managing a mandibular fracture was N89, 312.20 ($488) per person, and this corresponds to 8.4% of the health care finance of the state [43]. The scoping review aims to highlight the existing knowledge gap in the distribution, prevalence, incidence, risk factors, mortality, and economic burden of MI among adults in SSA. The outcome of the proposed scoping review will inform future research policy and healthcare decision-makers to effectively utilize the inadequate medical resources and develop more directed policies. These policies will align with the matching social development level, gender difference, and age distribution for the interventions on injury prevention.

## 2. Material and Methods

### Design

A scoping review systematically maps the literature on a topic by identifying key concepts, theories, and sources of evidence that inform practice in the field. Therefore, this projected scoping review will employ the available literature (peer-reviewed and grey) on the distribution of maxillofacial injury involving adults in SSA underpinning, prevalence, incidence, risk factors, mortality, and the economic burden. This review forms part of the main research study, which aims to define the burden of maxillofacial injury among adults in Lesotho. The database search and screening outcomes from diverse studies will be reported using the PRISMA flow diagram. The proposed scoping review will be reported by applying the guidelines for observational studies in epidemiology and the preferred reporting items for systematic reviews and meta-analysis extended for scoping reviews (PRISMA-ScR) checklist and explanation [45]. The methodological frameworks described by Arksey and O’Malley [46] and Levac [47] will be used for analysis. These frameworks will analyse the scoping review in six (6) steps; (i) identifying the research question, (ii) identifying the relevant studies, (iii) study selection, (iv) charting the collected data, (v) data collating, summarizing and reporting, and (vi) Consultation (optional).

Step 1: To identify the research questions.


*The principal research question of the scoping review is as follows:*


What is the prevailing evidence on the distribution of maxillofacial injury among adults in SSA?


*Sub-questions*


(a)What is the burden of maxillofacial injury in SSA with estimations on the prevalence, incidence and mortality?(b)What risk factors are associated with maxillofacial injury in SSA?(c)What are the estimated costs associated with maxillofacial injury?

The eligibility of the research question for this proposed scoping review will be determined using the; population, concept, and context (PCC) framework [48], as demonstrated in Table 1.

Step 2: Identification of relevant studies


*Sources of information*


The authors will conduct the keyword search of literature in English. In the proposed review, identifying the studies of relevance will be guided by the inclusion and exclusion criteria, which will be carried out without a date limit. The author will perform an electronic literature search through the following databases: Google Scholar, Web of Science, Medline, EBSCOhost, Cumulated Index to Nursing and Allied Health Literature (CINAHL), Health Source: Nursing/Academic Edition, PubMed, Embase, Emcare, and Scopus. Furthermore, we will search for grey literature from institutional repositories, government and international organisations’ reports such as the WHO and from university dissertations and theses. Foremost to be employed in the study will be a comprehensive search approach that considers the keywords, medical subject headings (MeSH), or subject headings search terms that relate to key concepts, as well as Boolean terms “AND” and “OR”. The keyword terms to be used for the literature search will include; maxillofacial trauma, maxillofacial injury, facial trauma, facial injury, facial bone trauma, facial bone injury, facial lacerations, mandibular fracture, mandibular trauma, mandibular injury, maxillary fracture, maxillary injury, nasal fracture, nasal injury, orbital fracture, orbital injury, epidemiology, prevalence, incidence, risk factors, disability, mortality, burden, comorbidities, associated cost and countries in Sub-Saharan Africa. The piloted search strategy in PubMed is demonstrated in Table (Table 2). Furthermore, the search for relevant articles will be obtained from the reference list of the included articles (snowball approach). To ascertain reliability between reviewers, a training workshop will be conducted before the screening. In case of discrepancies between the reviewers, a third arbitrator/reviewer will revisit the inclusion and exclusion criteria and institute a further pilot test. The compilation of all relevant articles and the identification of duplicate records will be achieved by employing the EndNote reference manager.

Step 3: Study selection and inclusion criteria

Eligibility criteria


*Inclusion criteria*


This proposed review will include all studies conducted among the adult population aged 18 years and above that present evidence on any of the factors listed below:The prevalence of maxillofacial injury in SSAThe incidence of maxillofacial injury in SSAThe risk factors associated with maxillofacial injury in SSAThe comorbidities associated the maxillofacial injuryStudies that have a clear definition of maxillofacial injury


*Exclusion criteria*


This review will exclude the following:Evident articles involving individuals under the age of 18 yearsReview studiesStudies conducted outside the setting of SSAClinical trials and intervention-based studiesStudies carried out in other languages other than English and do not have an English versionStudies that lack a clear definition of maxillofacial injury


*Selection Process*


The screening process of articles will be carried out in three stages, namely, title, abstract and full article screening. The principal investigator (PI) will conduct the title screening. In contrast, the abstract and the full article screening will be autonomously carried out by the PI and the co-reviewer (CR). A comprehensive title screening will be conducted in the electronic databases described by the PI, with the assistance of the Institution (UKZN) librarian. Articles found eligible at the title screening stage will be exported to an Endnote version 20 library, where duplicates will be detached. Two independent reviewers, the PI and CR, will co-screen articles for valuation against the inclusion criteria, and studies that meet the inclusion criteria will be incorporated for the next phase of abstract screening. This stage will be followed by full article screening, independently conducted by the PI and CR. These two (PI and CR) will appraise the selected articles following the inclusion criteria. In the case of any discrepancy in a selected article at both stages (abstract and full article) between the PI and CR, clarifications and discussions will be held to obtain a consensus. The failure to obtain a consensus between the PI and CR will result in the help of a third screener for resolution. The EndNote library “Find full text” option will be applied to download PDFs of exported studies spontaneously. The assistance of the Institution librarian will once more be sought at the full article stage on acquiring articles that cannot be fully accessed freely. Furthermore, the assistance of corresponding authors will be sought through email to request a full-text article, if essential. The level of agreement between screeners’ (interrater) results after screening for abstracts and full articles shall be determined through the application of Cohen’s kappa statistic on reliability [40]. The interrater result will be measured as; values ≤ 0 = no agreement, 0.01–0.20 = none to a slight agreement, 0.21–0.40 = fair agreement, 0.41–0.60 = moderate agreement, 0.61–0.80 = substantial agreement and 0.81–1.00 = perfect agreement. Any kappa below 0.60 will not be reported, as this indicates insufficient agreement among the raters, and insignificant confidence is awarded on such study results [49]. The screening result will be reported following the updated PRISMA flow diagram [50], as described in Figure 1.

Step 4: To chart the data.

The PI will employ the Google form for this process by developing a data charting (extraction) form. After thoroughly reading the full text, the PI will use the form to extract information from each relevant study. The pilot study of this process (data charting) will be conducted to guarantee its accuracy and carried out by the reviewers before the scoping review. The feedback received from this process will guide the modification of the data form. This process will be regularly updated during the entire period of the scoping review to allow the capturing of all pertinent data applicable to answer the research question. The details to be captured by the data form will include; (i) Author and date of publication, (ii) Study setting, (iii) Publication type, (iv) study design (sample size), (v) Peak age range of incidence, (vi) Male/Female ratio, (vii) Major cause of Maxillofacial injury, (viii) the second major cause of Maxillofacial injury, (ix) Maxillofacial soft tissue affected, (x) cost of management, (xi) other relevant findings to answer the research question, and (xii) conclusion.

Step 5: To collate, summarize and report the results.

This stage will encompass the thematic analysis of the extracted data, for reporting and making sense of the results. The extracted data from the included articles will be descriptively summarized and exposed to a thematic analysis by employing Nvivo version 12. The presentation will assume the narrative of the appropriate themes and sub-themes along with the research questions as the; (i) prevalence, incidence and mortality, (ii) risk factors, and (iii) associated costs. The use of tables and figures will be employed in the result presentation when necessary.

Step 6: Methodological quality appraisal

The quality of the methodological method employed will be determined by applying the mixed methods appraisal tool (MMAT) [51]. Two independent reviewers will carry out this process. The tool (MMAT) will evaluate the methodological quality of qualitative studies, quantitative descriptive studies, and mixed methods studies. The application comes with a set of questions that will assess the suitability of the different sections reported in each evidence source. The grading will determine the quality of evidence as percentage scores and described as; (i) ≤50% = low-quality evidence, (ii) 51–75% = average quality evidence, and (iii) 76–100% = high-quality evidence. However, studies rated as low-quality evidence will not be excluded, as this is discouraged by MMAT.

Ethical approval and consent to participate

This review will be exempted from ethical approval, for it is based on data reported in publicly accessible databases and publications.

Potential limitations to the scoping review

There is a paucity of literature on the epidemiology and the associated economic burden of maxillofacial injury in SSA. Hence, this may result in limited literature for the scoping review. Furthermore, there is the potential to miss relevant articles given that articles published in English will only be considered for the review.

## 3. Discussion

The proposed scoping review aims to map the available evidence regarding the epidemiology, risk factors, and estimated costs associated with maxillofacial injury among adults in sub-Saharan Africa. The review is expected to highlight the under-explored areas in this injury. This scoping review will comprise observational studies and cost of management study designs carried out among adults in SSA. All relevant studies in English, and without date limitations will be included. To be excluded from the review will be randomized controlled clinical trials and intervention-based studies. This is because the data from such methodologies will not be appropriate for addressing the research question of the coping review. The review is a pioneer study to map the evidence on the distribution of MI in SSA, with appraisals on prevalence, incidence, mortality, risk factors, and associated cost. The National Trauma Data Bank report revealed that approximately 25% of all recorded injuries involve the face [38]. The facial region is the most visible among the body parts, delicate in structure, and lacks protection [44,52], resulting in it being the leading cause of morbidity [32,43]. The high incidence and financial cost of treatment and rehabilitation and the possibility of the devastating irreversible damage that might result from a facial injury have made it a significant challenge for public healthcare services [7,15,53]. Hence, this invariably inflicts a substantial economic burden on the national health budgets, further constraining the present delicate health systems in SSA. Mapping out the burden of this injury over time is crucial in predicting the impending outcomes of this health-related event.

The sustainable development goal (SDG) 3 targets ensuring healthy lives and promoting well-being for all ages [54]. Hence, there must be a detailed public health approach study on all natures of injury to achieve this goal. On the contrary, in SSA, the knowledge of this nature of injury (MI), the associated risk factors, and costs remain limited compared to high-income countries (HICs). However, this information is critical to inform health strategies for improving health outcomes and injury prevention. The scoping review will generate evidence from the present literature to reveal gaps in research and guide the methodology of the main study. The data extracted will be demonstrated in a tabular form to ensure that the aims of the scoping protocol are fulfilled. The tabulated results will accompany a descriptive summary showing the relationship between the protocol’s aim and research questions. The strengths and limitations of the methodology will be presented in detail, thereby ensuring the study’s credibility.

## 4. Conclusions

The outcomes from this projected scoping review are anticipated to give a projection on the distribution of MI in SSA with appraisals on prevalence, incidence, risk factors, mortality, and associated economic burden. Hence, the synthesized evidence from the study will help researchers, decision-makers, and other stakeholders to inform policy and ensure an effective allocation of healthcare funding. Furthermore, this will help in upgrading the healthcare system performance and thus improve measures on injury prevention and the interventional process.

## Figures and Tables

**Figure 1 ijerph-20-01531-f001:**
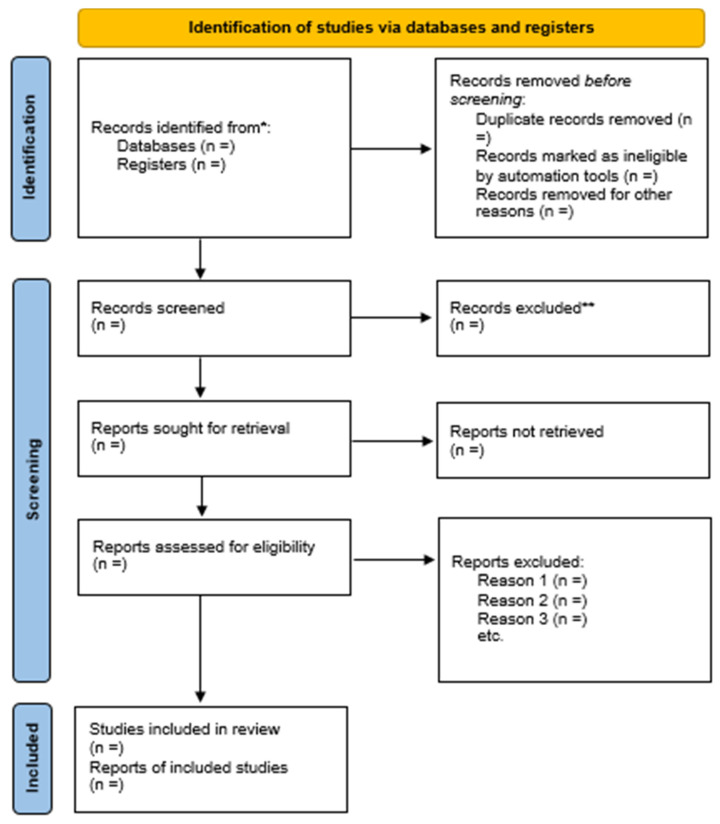
PRISMA flow diagram of the study selection process. PRISMA, Preferred Reporting Items for Systematic Reviews and Meta-Analyses. * Consider, if feasible to do so, reporting the number of records identified from each database or register searched (rather than the total number across all databases/registers). ** If automation tools were used, indicate how many records were excluded by a human and how many were excluded by automation tools. From: Page, M.J., McKenzie, J.E., Bossuyt, P.M., Boutron, I., Hoffmann, T.C., Mulrow, C.D., et al. The PRISMA 2020 statement: an updated guideline for reporting systematic reviews. BMJ 2021; 372:n71. doi: 10.1136/bmj.n71. For more information, visit: http://www.prisma-statement.org/ (accessed on 11 October 2022).

**Table 1 ijerph-20-01531-t001:** PCC framework for defining the eligibility of the studies for the principal research question.

Population	Adult, 18 Years and above with Maxillofacial Injury
Concept	Maxillofacial injury
Context	Countries in sub-Saharan Africa

**Table 2 ijerph-20-01531-t002:** Pilot search in PubMed electronic database.

Population	Concept	Context	Key Words	Date	Number Found
Adult, 18 years and above with maxillofacial injury	Maxillofacial injury	Countries in sub-Saharan Africa	((((((((maxillofacial injury [MeSH Terms]) OR (facial injury [MeSH Terms])) AND (epidemiology [MeSH Terms])) OR (incidence [MeSH Terms])) OR (prevalence [MeSH Terms])) AND (risk factors [MeSH Terms])) AND (cost [MeSH Terms])) OR (cost analysis [MeSH Terms])) AND (sub-Saharan Africa [MeSH Terms])	1 June 2022	2796

## Data Availability

The datasets used and analyzed during the scoping review will be obtainable from the corresponding author upon reasonable request.

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
