# Peer review of "Mapping Evidence on the Epidemiology and Cost Associated with Maxillofacial Injury among Adults in Sub-Saharan Africa: A Scoping Review Protocol"

_ijerph, 2023, doi:10.3390/ijerph20021531_

Round 1

Reviewer 1 Report

Dear Authors, first of all I would like to congratulate You on your work. The topic Is of great clinical relevance. However, I believe that the article could be improved. Please, take a note of some suggestions.

The abstract needs to be re-structured by shortening the methodology and adding highlighted results and a clear conclusion/ outcome of the study.

Introduction- this section is short and should be more focused on the topic in question. If possible describe more details about Maxillofacial Injury , I would suggest these papers:  Maxillofacial and concomitant serious injuries: An eight-year single center experience DOI 10.1016/j.cjtee.2016.11.003.

Oral and maxillofacial emergencies: A retrospective study of 5220 cases in West China. Dent Traumatol. 2022 Nov 11. doi: 10.1111/edt.12798.

Materials and methods- this section is well organized, 

Please, describe this section  meticulously as this is very important for the readers. 

 Statistical analysis is very basic.

I believe that your manuscript would have much more relevance after suggested improvements.

Author Response

  • The result session was added in the abstract as advised by reviewer 1 (highlighted) and this was accomplished maintaining the number of words for abstract as prescribed in the author’s guidelines.
  • More details on maxillofacial injury was described and highlighted in introduction session of the manuscript (highlighted on pg 1 of 9 & 2 of 9).
  • This section was described in full as prescribed by presented by Arksey and O’Malley and extended by Levac and colleagues. The result comes as thematic analysis of the data extracted (not statistical).

Reviewer 2 Report

The article represents a study design and planning description focused on maxillofacial injuries among adults in Sub-Saharan Africa. Despite the rising importance of the clinical problem, it has to be concluded, that the study design itself doesn't contain any novel data or outlooks. Thus, study designs or protocols (except some special cases) should not be considered as material for publication in high-rated scientific journals.

However, it could be suitable as a preprint publication.

Author Response

This is a protocol designed to identify the distribution of MI in SSA. Despite the rising incidence globally, this nature of injury has not been given the required attention proportionate to the epidemiology, especially in the sub-Saharan Africa. Therefore the synthesized evidence from this protocol will create more awareness and encourage researchers into exploring further, and identifying further gaps into maxillofacial injury in this region (SSA).

Reviewer 3 Report

The Reviewer would like to thank the authors for their efforts in writing the manuscript and for publishing the protocol of the study.

Would you please remove only the discussion and conclusion section from the manuscript as it's still so early to discuss the results or drive conclusions.

Author Response

Thanks for the comments. Please note that the discussion portion was not about the result obtained, but rather on the proposed scoping review and the importance.  This is fully explained in the statement from page 6 of 9, “The review is expected to highlight the under-explored areas in this injury. This scoping review will comprise observational studies and cost of management study designs carried out among adults in SSA. All relevant studies will be included, with neither language nor date limitations……”

The conclusion, gave a projection of the outcomes expected from the scoping review when carried out. This is further explained on pg 7 of 9.

Thanks.

Reviewer 4 Report

The authors of the publication submitted for review are Adekunle Adeleke, Mbuzeleni Hlongwa, Sizwe Makhunga and Themba G. Ginindza. Among them, there are specialists in epidemiology and general dentistry as well as public health managers. Unfortunately the absence of a maxillofacial surgeon in this team is very noticeable, due to the fact that the terminology and technical language directly associated with maxillofacial injury in the reviewed text is not compatible with standardized nomenclature used in professional literature.

The manuscript that I received for reviewing is a scoping review protocol concerning publications on the subject of epidemiology and cost of treatment for adult patients admitted to hospitals in Sub-Saharan Africa with injuries to their maxillofacial region.

The authors are briefly mentioning the aim of this work in the text of the manuscript, however, it is not very legible and needs to be redrafted. The potential reader needs to be clearly informed what exactly is the purpose of this publication and what type of questions did the authors want to answer in a scoping review that was constructed in this way.

What also needs to be addressed by the authors is the fact that they did not attempt to write a systematic literature review protocol and instead focused solely on a scoping review protocol. Why was such a decision made?

Also, a very concerning issue to me is relying on Google Translate in order to include articles that were not originally written in English. In my opinion this is not a sufficient tool and such an approach is burdened with a high risk of error. Not even the best on-line translator (which in my humble opinion would be DeepL and not Google Translate) can be used in case of any scientific and academic literature transcription but especially medical papers.

An explanation needs to be made as to why a well-renowned and recognized databases such as EBSCO, Embase or Emcare haven’t been used for the literature search. The authors also have to clarify as to why the grey literature has been included in the article?

There’s a lot of typos in the text, double spaces, repetitions, grammatical errors, etc. It is absolutely crucial for the manuscript to be thoroughly re-read and all the linguistic errors have to be carefully corrected, preferably by a professional translator or a native speaker.

The Fig. 1 on page 5 is not clearly visible and has to be readjusted.

In the current form the text requires extensive modifications and considerable redrafting and is not eligible for publication. I want to give the authors a chance though, so I qualify it as one that needs a major revision and I really hope to see the manuscript submitted for a second round of reviewing only after substantial changes are made to it.

Author Response

Thanks for the comments.

The response to comments;

  • Extensive editing of English language and style done with the assistance of language (English) expert.
    • Please, note that the PI has been in the maxillofacial clinic as a Registrar from 2002 till 2015 and currently lectures at the National Health Training College on three modules; i, Oral surgery, ii, Clinical dentistry and iii, Ethics and legal issues in dentistry from 2016 till present. Hence, the terms used and technicality were linked to maxillofacial injuries in the manuscript, a major area that is within the PI scope of work. Furthermore, the Scoping protocol and the review will focus on the epidemiology and burden of MI and not the treatment.
    • On Research question:This was stated on page 2 of 9. The eligibility of which was determined using the; population, concept and context (PCC) framework.
    • Why scoping review protocol? This was because the Protocol will guide in mapping the literature on maxillofacial injuries epidemiology. Furthermore, it will guide the scoping review in identifying gaps in available evidence, objectives and methodology of subsequent research in this nature of injury.
    • Concerns on Google translate: This was reconsidered, and changed to language expert (pg 3 of 9, the first paragraph). Furthermore, the only study identified in SSA that was in French was published with the English translation.
  • On the database searched: Revealed on page 3 of 9 of the manuscript is a pilot search, where PubMed was employed. The manuscript gave a full description on how this will be carried out. To be employed in database search will include; Google Scholar, Web of Science, Medline, Cumulated Index to Nursing and Allied Health Literature (CINAHL), Health Source: Nursing/Academic Edition, PubMed, Scopus, World Health Organization (WHO) library databases and grey systematic search. This was explained and highlighted on page 3 of 9.

Round 2

Reviewer 4 Report

It is with utmost regret but the authors of this manuscript simply give me no choice other than to reject this publication. I am not satisfied with their response to my comments and the updated version of the text is not that much different from the original despite the fact that I suggested major revision. The authors have not taken my remarks seriously - all of the adjustments that were made to the text have been very superficial and did not go in accordance with my comments, some were totally ignored, like the need for clarifying the aim of this work or correcting Fig. 1 which is not visible enough. Changing the use of Google Translate to an unspecified “language expert” leaves me with a great deal of doubt as to whether such a person was even considered in this study in the first place. In my opinion this change has been made purely to satisfy the reviewer and does not reflect reality. 

There needs to be a clear understanding on the authors’ side that cooperation with reviewers and editors is a crucial part of the publishing process and cannot be treated lightly. It is not in our interest nor it is our ill will to reject articles without valid reasons. On the contrary, the suggestions that all reviewers provide are there to aid the authors and help them improve their work, not cause frustration or feelings of injustice. Nonetheless, this paper is not acceptable in its current form and I think that enough time was given to the authors to make necessary improvements, yet they didn't utilize that time effectively.

Author Response

Dear Reviewer (4),

We thank you for taking time to review this paper. We are grateful for the comments aimed at improving the quality of this work. We believe that we have taken all reviewers’ comments seriously and have incorporated their recommendations with the revised version of the manuscript.

We would appreciate it if you would be a bit more detailed as to which sections were not responded to in full.

Please find below the consolidated responses to your comments.

1) The reviewer asked about the purpose of this review and the questions we aim to answer.

Response

  1. Please note that the type of questions used in this scoping review have been highlighted in lines 86 – 95 (evident and highlighted on page 2 of 9, step 1).
    1. Our principal research question of the scoping review is: What is the prevailing evidence on the distribution of maxillofacial injury among adults in SSA?
    2. Sub-questions are:
      1. What is the burden of maxillofacial injury in SSA with estimations on the prevalence, incidence and mortality?
      2. What risk factors are associated with maxillofacial injury in SSA?
      3. What are the estimated costs associated with maxillofacial injury?

             c. We have also indicated the purpose of this paper in the introduction section that it aims to highlight the existing knowledge gap in the distribution, prevalence, incidence, risk factors, mortality, and economic burden of MI among adults in SSA.

2. On the absence of maxillofacial surgeon in team.

Response;

  1. Please, note that;
    1. The PI has been in the maxillofacial clinic as a registrar from 2002 till 2015 and currently lectures at the National Health Training College on three modules; i, Oral surgery, ii, Clinical dentistry and iii, Ethics and legal issues in dentistry from 2016 till present. Hence, the terms used and technicality were linked to maxillofacial injuries in the manuscript, a major area that is within the PI scope of work.
    2. Furthermore, the Scoping protocol is on the epidemiology and burden of MI and not the treatment.

3. The reviewer also asked why the authors did not attempt to write a systematic literature review protocol and instead focused solely on a scoping review protocol.

Response;

We believe that scoping review and systematic reviews serve a different purpose. Ours is to map evidence on the Epidemiology and Cost Associated with Maxillofacial Injury among Adults in Sub-Saharan Africa. As a result, we believe that conducting a scoping review is more appropriate because it is able to systematically map the literature on a topic by identifying key concepts, theories and sources of evidence that inform practice in the field. We believe that scoping reviews are an ideal tool to determine the scope or coverage of a body of literature in a particular topic (evident and highlighted on page 2 of 9).

4. The reviewer suggested additional of other databases such as, Embase, EBSCOhost and Emcare

Response;

In terms of using EBSCO or Embase, and Emcare databases, we have now included these databases and we will search them for relevant articles (evident and highlighted on page 3 of 9, step 2).

5. On the use of Google Translate in order to include articles that were not originally written in English.

Response;

The authors share the same sentiments with the reviewer that this will likely lead to high risk of error. Therefore, we have decided to indicate that we will only include articles that are published in English. As a result, we have noted this in our limitations section (evident and highlighted on pages 3 of 9 and 4 of 9).

6. Clarifications on the inclusion of grey literature in the article.

Response;

The authors are of the opinion that the inclusion of grey literature from institutional repositories, university dissertations and theses will ensure that every source of detailed information is captured to give a comprehensive review (evident and highlighted on page 3 of 9).

7. Editing figure one for visible picture

Response;

Done as advised (evident on page 5 of 9)

8. On spaces, repetitions, and  grammatical errors

Response

Editing done by a Proof-reader.

Thanks and regards,

Adeleke Adekunle (Dr)

Round 3

Reviewer 4 Report

The authors finally took the reviewer's comments seriously and made most of the necessary improvements. But literature needs to be urgently updated. There are no works/articles from the year 2022 and only 2 from the year 2021. Taking into consideration that the paper talks about a very new subject, this has to be corrected and more recent publications have to be included. 

Author Response

Dear Reviewer,

I trust that you are well. This is the response to the last comment shared.

We once again thank the 4th reviewer for taking time to review this paper. We are grateful for the comment aimed at improving the quality of this work. We believe that we have taken the reviewer’s comment seriously and have incorporated the recommendations with the revised version of the manuscript.

Please note that literature is now updated as advised, and contains the available latest articles relevant for the study. This is evident in the reference session of the manuscript and tracked.

Thanks and regards,

Adeleke Adekunle (Dr)
